# The Efficiency of Pest Control Options against Two Major Sweet Corn Ear Pests in China

**DOI:** 10.3390/insects14120929

**Published:** 2023-12-06

**Authors:** Xin Li, Yanqi Liu, Zhichao Pei, Guoxiang Tong, Jin Yue, Jin Li, Wenting Dai, Huizhong Xu, Dongliang Shang, Liping Ban

**Affiliations:** 1Department of Grassland Resource and Ecology, College of Grass Science and Technology, China Agricultural University, Beijing 100193, China; lix449129@gmail.com (X.L.); bs20203240989@alu.cau.edu.cn (Y.L.); lijin2017@163.com (J.L.); daiwenting0513@163.com (W.D.); xhz_6989@163.com (H.X.); sdl18810122231@sina.com (D.S.); 2Beijing Agricultural Technology Extension Station, Beijing 100193, China; pzc2010@163.com; 3Beijing Fangshan District Planting Technology Extension Station, Beijing 102499, China; nkstgz@163.com; 4Beijing Plant Protection Station, Beijing 100193, China; yuejin_612@163.com

**Keywords:** corn phenological stage, *Helicoverpa armigera*, *Ostrinia furnacalis*, synthetic insecticides, biopesticides

## Abstract

**Simple Summary:**

Corn ear pests, such as cotton bollworm and corn borer larvae, can infest corn ears within hours to a day, posing a significant challenge for their control. This study aimed to determine the ideal timing for pesticide application, select effective and safe pesticides, and optimize application methods. Over several years, we monitored adult sweet corn ear pests in Beijing, investigating their relationship with corn phenology and local climate to identify the best time for effective pesticide application. Through field experiments evaluating 13 different pesticides, we identified cost-effective and safe options. Ultimately, we discovered that in mid-June, approximately five days after corn silking in sweet corn fields, moth populations got the greatest growth rate. During this crucial period, applying Indoxacarb + Chlorfenapyr, Virtako + mineral oil, or the biopesticide *Beauveria bassiana* + oil on corn silks and ear surfaces maximized their effectiveness. Our research significantly improves corn ear pest control.

**Abstract:**

*Helicoverpa armigera* (Hübner) and *Ostrinia furnacalis* (Guenée) are the most devastating insect pests at the ear stage of maize, causing significant losses to the sweet corn industry. Pesticide control primarily relies on spraying during the flowering stage, but the effectiveness is inconsistent since larvae are beneath husks within hours to a day, making pesticide treatments simpler to avoid. Insufficient understanding of pest activity patterns impedes precise and efficient pesticide control. *H. armigera* and *O. furnacalis* in corn fields were monitored in the last few years in Beijing China, and we observed a higher occurrence of both moths during the R1 stage of sweet corn. Moth captures reached the maximum during this stage, with 555–765 moths per hectare corn field daily. The control efficiency of nine synthetic insecticides and five biopesticides was assessed in the field during this period. Virtako, with mineral oil as the adjuvant, appeared to be the most effective synthetic insecticide, with the efficiencies reaching 88% and 87% on sweet and waxy corn, respectively. Pesticide residue data indicated that the corn is safe after 17 days of its use. The most effective bioinsecticide was *Beauveria bassiana* combined with mineral oil, with 88% and 80% control efficiency in sweet and waxy corn, respectively. These results suggested that spraying effective insecticides 5 days after corn silking could effectively control corn ear pests *H. armigera* and *O. furnacalis*. Our findings provide valuable insights for the development of ear pest management strategies in sweet corn.

## 1. Introduction

Sweet corn (*Zea mays convar. saccharata var. rugosa*) is an important crop cultivated worldwide. Demand for sweet corn, including sweet and waxy corn cultivars, has been expanding in recent years, accounting for nearly 18% of the overall value of maize output [1,2,3]. However, sweet corn, especially its ear, is known to be susceptible to phytophagous insects in the summer, and the same insect species feeding on conventional grain corn can be found in sweet corn. Among the sweet corn insect pests, most are moths from the order of Lepidoptera, including Asia corn borer *Ostrinia furnacalis*, cotton bollworm *Helicoverpa armigera*, beet armyworm *Spodoptera exigua*, and peach pyralid moth *Conogethes punctiferalis*, the first two species were found as the most devastating corn ear pests in China and Southeast Asia. These two pests have caused tremendous damage to the sweet corn industry equivalent to a $1.3 billion loss each year in China [3,4].

Corn ear pests initially feed on the silk, but then quickly attack the grain during the corn milk stage and live inside the ear for the rest of their larval period, which protects them from chemical spraying. Therefore, the development of a highly efficient chemical-based corn ear pest control strategy is needed. Pesticide applications on the silk rather than leaves and the entire plant were found as the more effective and environmentally friendly control method to reduce damage caused by corn earworms [5,6]. In addition, insecticide application time is a critical factor for pest control because the occurrence of each pest follows its own time pattern, and following these patterns can maximize control efficiency [7]. The larvae of *H. armigera* and *O. furnacalis*, the two insect pests studied here, burrow into corn grains through cornsilk once hatched and then live beneath husks until pupation, making them less exposed to pesticide application. Thus, spraying insecticides before larvae burrow is crucial for the management of these two corn pests.

The timing of insecticide application should be based on the pest occurrence, but it is important to note that lepidopteran pests are highly sensitive to environmental factors. Temperature and humidity are two major factors that can influence lepidopteran pest occurrence, and therefore, are typically included in pest forecasting systems to improve the precision and effectiveness of pest control [8,9]. Climatic conditions have influenced the fecundity of insects, and the corresponding pest prediction model was established to investigate the occurrence of Lepidopterans based on the number of eggs laid and meteorological data [10]. However, the statistics data may contain large random errors and it will decrease the accuracy of the model because Lepidopterans often lay their eggs in hidden parts of the host plant and other plants nearby. Interestingly, studies have found non-environmental factors such as plant volatiles, plant variety, and inherent insect pheromones, etc., are the variables attractive elements that can exert their impact on herbivore movement [11,12], and those are highly related to the phenological period of the plant. Pest prediction and control should take those factors into consideration as well. Sex pheromones (chemotaxis) and phototaxis have been extensively used for insect population monitoring and management in cornfields. This is particularly valuable for predicting Lepidopteran moth activity, enabling precise timing of insecticide application [13]. Studies have found that the corn borer usually appears dominantly near the period of maize pollination [14].

Entomologists have long sought accurate and effective pest control methods—not only to maximize crop yield through pest management—but also to manage pests in a sustainable way that causes the least hazards to the environment including beneficial insects. To date, synthetic insecticides are still the most common and effective compounds of reagents used in corn pest management, with almost >80% control efficiency in most cases. However, synthetic insecticide residues have become a great concern, particularly for sweet corn, which human ingests directly, and therefore, selection of insecticides with low human toxicity is required. While biologicals are considered to be safer than synthetic insecticides, they are not generally as efficient as synthetic chemicals, which limits their application in corn fields [15]. An exception is the use of *Trichogramma*, a genus of parasitoid wasps, as natural enemies that showed a high control efficiency on ear pests in cornfields [16]. By employing agricultural unmanned aerial vehicle intelligence to deliver parasitic wasp eggs to control corn borer in cornfields, the average control efficiency reached 84% [17]. However, sweet corn is generally seeded during a one-week interval, and results in a surge of newly-laid lepidopteran eggs continually in the field, which necessitates parasitic wasp release weekly. Frequent usage will considerably increase the cost. Furthermore, the parasitic wasps exhibit a preference for specific host species, and this largely limits their usage in corn fields because there are many ear pest species found in the fields. Another efficient biological control approach is to employ *Beauveria bassiana*, a fungus pathogen, to control storage corn straw pests; however, its efficiency is low when used for ear pest control [18].

Our hypothesis is that corn ear pest infestation activity is related to maize physiological phase difference and local meteorological conditions during the earing stage. Our objectives are to find and use infestation patterns in early summer, as well as conduct efficient pesticide selection studies, to build a more effective corn management approach. To optimize sweet corn pest management, in this study, we investigated the populations of corn ear pests and their relationship with the corn phenological period and with meteorological data. We also compared the control efficiency and costs in the field among several insecticides, including both synthetic and biological agents, tested the persistence of residues, and further assessed the improvement of control efficiency when insecticides were applied with adjuvants. Our findings demonstrated that spraying insecticides precisely in a single dose during the key phenological period could effectively control corn ear pests in Beijing, China.

## 2. Methods

### 2.1. Pests Population Monitoring

Sex pheromone trapping: The experiment was carried out in the corn experiment station in Fangshan district, Beijing, China (E115°99 N39°65) in 2018–2019. Sweet corn variety Jingketian183 and wax corn variety Jingkenuo2010 were planted at a planting density of 55,556 plants/ha on 3 May 2018 and 5 May 2019. The planting strategy consisted of alternating 667 m^2^ of sweet corn with 0.0667 hectares of wax corn, with each monitoring treatment group occupying 667 m^2^ of corn (half sweet corn and half wax corn). A protective perimeter of four rows of maize was placed around each treatment group (*n* = 3). The pheromone traps (Pherobio Technology Co., Ltd., Beijing, China) were set up to monitor the populations of five lepidopterans, which are *Helicoverpa armigera*, *Ostrinia furnacalis*, *Spodoptera exigua*, *Dichocrocis punctiferalis*, and *Mythimna separata*. The sex pheromone of each moth was placed in delta traps, and four traps were randomly distributed in every treatment group area for each pest, and the distance between the bottom of the delta traps and the ground was 1.2 m. Monitoring of these species started after corn sowing until corn harvest. Insect collection, and species counting and identification were conducted every three days. The traps were reset every seven days.

Light trapping: Sweet corn variety Jingketian183 and wax corn variety Jingkenuo2010 were planted as well at the same density on 2 May 2019. The planting strategy consisted of alternating 667 m^2^ hectares of sweet corn with 667 m^2^ hectares of wax corn, with each monitoring site occupying 0.335 hectares of corn (half sweet corn and half sticky corn). Moths were collected using a frequency-vibration insect-killing lamp (YF-TY-40, Henan Yunfei Technology Co., Ltd., Zhengzhou, China) in three different monitoring sites (as three replicates). Site A (E116°83, N39°68), site B (E115°86, N40°48), and site C (E115°99, N39°65), were set up at over 30 km from one another in Beijing, China, with one light trap at each site. The lamp has a coverage capacity of 1.334-hectare corn field based on the instruction manual. We started collecting on 1 June 2023, 30 days after the corn sowing, until corn harvest, (45 days in total) daily from 8:00 PM to 6:00 AM, and the moths were determined and quantified after collection every day.

### 2.2. Insecticide Test

Jingketian183 and Jingkenuo2010, used for insecticide testing, were planted in 2019 (E115°99 N39°65) and 2021 (E115°99, N39°65) in the same way at the same time compared with moth monitoring. Each pesticide treatment for each corn variety involved 333 m^2^ of corn (1800 plants), with 150 corn plants forming one replicate. Three replicates and three control groups were established and randomly distributed from each other. Two rows of corn were placed between treatment groups as buffer rows.

In 2019, Control efficiency for pest management at different phenological stages was assessed using four synthetic insecticides. Fourteen insecticides (Table 1) were tested on sweet corn, including nine synthetic and five biological insecticides. The synthetic insecticides were Deltamethrin, Fenitrothion, Phoxim, Chlorantraniliprole, Indoxacarb, Indoxacarb + Chlorfenapyr, Beta-cypermethrin, Virtako, and Cyfluthrin, and the bioinsecticides were *Beauveria bassiana*, *Bacillus Thuringiensis* (Bt), Azadirachtin, Matrine, and Veratrine. The decreasing efficiency rule of insecticides (*B. bassiana* vs. *B. bassiana +* mineral oil; Virtako vs. Virtako + mineral oil) was examined in 2021.

Insecticides were diluted with 30 L of water to apply in 667 m^2^ of corn fields, while the control group received 30 L of water only. In the V12 stage of corn, the pesticide solution is evenly sprayed on the leaves and stems of the plants. During the silking stage, the pesticide is only sprayed on the corn filament and the entire surface of the ear; 200 mL of emulsifiable mineral oil was added as an adjuvant reagent in some treatments. The insecticides were applied by using an electric sprayer (3WBD, Lan Yi Technology Co., Ltd., Shenzhen, China).

Corn damage was examined on the 0, 5th, 10th, 15th, and 20th days after insecticide application: immature corn plants with delayed growth were discarded, and the total number of usable corn plants was recorded. Observing the silk for the presence of live insects, insect excreta, and signs of feeding damage, such as bitten silk. Any visible holes on the surface of the corn ear were considered infested. In the last assessment, we manually opened the husk of the corn ear to assess internal infestation. The number of infested plants in each experimental group was documented. The corn damage rate and control efficiency were calculated using the following formulas:Corn damage rate (%) = (total number of damaged corn/total number of corn) × 100%
Control efficiency (%) = [(corn damage rate in treatment)/corn damage rate in control] × 100%

### 2.3. Insecticide Residue Analysis

These chemicals were used to test insecticide residues: ammonium acetate and sodium chloride (Sinopharm Chemical Reagent Co., Ltd., Beijing, China), acetonitrile and methanol (Thermo Fisher Scientific, Brussels, Belgium), Cleanert Florisil, Cleanert SC18, and Cleanert PSA (Agola Technologies, Hong Kong), and Chlorantraniliprole and thiamethoxam (purities ≥ 95%, Sigma-Aldrich, Missouri, USA).

Sample treatment and standard curve: In 2019, sweet and wax corn underwent ten different chemical treatments, resulting in a total of twenty treatment groups. Following the final control effect investigation, we collected three corn ears and three control samples for pesticide residue testing. Pharmacodynamic decrease regularity of Virtako was carried out in 2021, 60 corn plants for each treatment, 60 for the control group after spraying, three corn ears were collected on the 3rd, 4th, 5th, 6th, 7th, 10th, 15th, and 20th days and stored at −80 degrees Celsius for subsequent analysis.

The corn seeds in all treatments and the control groups were homogenized, and 2.0 g aliquot corn homogenate was mixed with 4.0 mL of acetonitrile and 0.3 mL of ddH_2_O, and vortexed for two minutes, which was followed by adding 0.4 g NaCl, vortexing for one minute, and then centrifuging for one minute at 4000 rpm. The supernatant (1.0 mL) extract was placed in a 10 mL centrifuge tube, and 0.1 g of SC18 purifier, 0.05 g of PSA purifier, and 0.05 g of Florisil purifier were added into the centrifuge tube to vortex for one minute. The supernatant mixture of each treatment was filtered through a 0.22 μm nylon filter membrane to obtain the corn matrix solution. Chlorantraniliprole and thiamethoxam were diluted with methanol to generate a standard compound (100 mg L^−1^). Corn matrix solution from the control group was used to dilute the standard compound into 10 mg L^−1^, 1 mg L^−1^, 0.5 mg L^−1^, 0.05 mg L^−1^, 0.01 mg L^−1^, and 0.002 mg L^−1^ of solutions, which were then analyzed in a LCMS-8050 system. A standard curve was generated by setting the nominal concentration (mg L^−1^) as the abscissa (X) and the peak area as the ordinate (Y). The corn matrix solution from each insecticide treatment was then detected.

Chromatographic and mass spectrometry conditions: An LCMS-8050 coupled with a triple quadrupole mass spectrometer (Scion TQ MS/MS, Bruker Daltonics, Los Angeles, USA) equipped with a Shim-pack GIST (2.1 × 50 mm internal diameter, SHIMADZU) was used to quantify the insecticide compounds in the corn seeds samples. Chromatographic conditions were as follows: mobile phases are 5 mmol L^−1^ ammonium acetate aqueous solution (A) with methanol (B), 0.25 mL min^−1^ of the flow rate, 2μL of the sample size, and 35 °C of the column temperature. The elution procedure involves 10% B for 0.00–0.01 min, 10% B for 0.01–1.00 min, 50% B for 1.00–2.00 min, 90% B for 2.00–2.01 min, 90% B for 2.01–6.00 min, 50% B for 6.00–6.50 min, 10% B for 6.50–6.51 min, 10% B for 6.51–8.00 min. The parameters for the mass spectrometry program are as follows: MRM in conjunction with SIM scanning mode was employed for data collecting in both positive and negative ion modalities. 3 L min^−1^ of the dry gas flow rate, 200 °C of the DL temperature, 2 L min^−1^ of the atomization gas flow rate, 3 L min^−1^ of the heating gas flow rate, 120 °C of the ion transmission tube temperature with 300 °C of the heating plate temperature.

### 2.4. Statistical Analysis

To compare the control efficiencies of sweet corn at two different phenological stages we conducted an independent-sample T test using IBM SPSS Statistics 20. The factors in our analysis are treatment time (spraying at V12 and 5 days after silking), while the control effect is considered the variable of interest.

We compared the control efficiency of both synthetic and biopesticides in sweet and waxy corn using one-way ANOVA with the Bonferroni–Holm post hoc test, conducted through IBM SPSS Statistics 20. To ensure that our data met the assumptions required for ANOVA, the Shapiro–Wilk test was used to assess normality and Levene’s test to examine homoscedasticity. Pesticide types were treated as factors, and control effects were considered variables of interest.

### 2.5. Meteorological Data

All temperature and humidity data of experimental plots were from the website: http://www.wheata.cn/, accessed on 1 May 2022. The weather stations were located in Shunyi District, Beijing, China (E116.585 N40.08), 28–64 km away from all tree experimental fields.

## 3. Results

### 3.1. Dynamic Occurrence of Predominant Pests in Ear Stage of Sweet Corn

We monitored the occurrence of the main pests, including *O. furnacalis*, *H. armigera*, *S. exigua*, *D. punctiferalis*, and *M. separata* in the field using sex pheromone traps during the growing period of sweet corn in 2018–2019. We found that *H. armigera* was the most predominant species in the field during the silking stage of corn, contributing over 50% of pests trapped; *O. furnacalis* was the second most predominant, accounting for approximately 30% of total pests trapped (Figure 1a). To mitigate potential biases arising from the singular use of a sexual attractant method and a single experimental location, in the third year, 2021, frequency-vibration insect killing lamps were used to monitor moths in the sweet corn field in three different monitoring sites, and the results were consistent with those from 2018 to 2019; *H. armigera* and *O. furnacalis* were still the most prevalent pests in corn fields, accounting for 67% and 21% of all moths. respectively (Figure 1a). When comparing the pest occurrence with the corn phenology, we found that the prevalence of the major pests, mainly the moth species, correlated to the phenological period of sweet corn. Moths were initially detected in sweet corn during the V12 stage, approximately 38–40 days after sowing, typically around mid-June in the Beijing area. Their density was observed to be 150 moths per hectare per day. Subsequently, the moth population experienced a significant surge during the silking and pollination period (R1), which occurred around 50 days after corn seed sowing, and then, reached its peak, 555–765 moths per hectare, approximately five days after R1 (Figure 1b). The temperature and humidity data of corn fields seemed to further explain the cause of the moth activity pattern. The average relative humidity swiftly changed to 75% in the V12 stage of sweet corn, a stage suitable for pest reproduction, and after that stage, the humidity remained at 60–96% until the corn reproductive growth stage (R2) (Appendix A) with suitable temperature and humidity for the occurrence of lepidopterans [19,20]. The average relative humidity and the temperature remained suitable throughout the silking phase and after. We consider the interval between the maximum pest development rate and the peak of pest incidence, from the start of corn silking to the end of corn pollination, nearly five days after silking, might be the optimal period for pest control.

### 3.2. Insecticide Application Time Exerted a Significant Influence on Control Efficiency

To determine if the silking period (five days after silking) is the optimal period for pest control, four widely used insecticides, Virtako, IndoxacarbIndoxacarb, IndoxacarbIndoxacarb + Chlorfenapyr, and Beta-cypermethrin, were applied once during the V12 stage or the silking stage, respectively, in 2019. The control efficiency was investigated around 20 days after silking period control, usually in the milk-ripe stage of corn ready to be harvested. The control efficiency for all four synthetics in the treatment during the silking period is significantly higher than that of the V12 stage insecticide spraying. Indoxacarb + Chlorfenapyr and Virtaco exhibited the highest control efficiency. The Indoxacarb + Chlorfenapyr treatment, applied with one spray at 5 days after silking, showed a 63% and 51% increase in control efficacy on sweet and wax corn compared to spraying at the V12 stage. Virtaco, on the other hand, increased control efficacy by 42% and 40%, respectively (Figure 2a). The remaining two pesticides, Indoxacarb and Beta-cypermethrin, showed lower control efficacy. However, in all groups, control efficacy was higher when pesticides were applied after corn silking compared to spraying at the V12 growth stage, resulting in a control efficiency increase of over 31% for all groups at least. It indicated that applying the pesticide five days after silking is effective for ear pest control, with significantly better results than at the V12 stage.

### 3.3. Screening of High-Efficiency Synthetic Insecticides and Safety Check

In order to identify the effective and low-toxic synthetic insecticides, nine insecticides commonly used by farmers were further utilized in sweet corn pest control studies in the field, in 2019. After 20 days of the pesticides being used. The results show that the Virtako + mineral oil was the most efficient treatment, with the control efficiency above 85% for both sweet and waxy corn (Figure 2b). Virtako without mineral oil showed a decreased control efficiency of 8% and 11% on sweet and waxy corn. The second was Indoxacarb + Chlorfenapyr, which had a control efficiency of 81% and 67% on sweet and waxy corn, respectively. Both Chlorantraniliprole and Indoxacarb showed control efficiency beyond 60%, with 66% and 67% for Chlorantraniliprole on sweet and waxy corn, and 62% and 60% for Indoxacarb (Figure 2b). However, for the remaining five insecticides, i.e., Deltamethrin, Fenitrothion, Phoxim, Beta-cypermethrin, and Cyfluthrin, the control efficiency was all less than 60% (Figure 2b). Among all these pesticides tested, four synthetic insecticides, Virtako, Indoxacarb + Chlorfenapyr, Chlorantraniliprole, and Indoxacarb were the most effective in controlling the ear pests of sweet and waxy corn. To ensure the safety of the nine synthetic insecticides, the amount of insecticide residue in sweet corn was tested by liquid chromatography-tandem mass spectrometry after the last investigation of the control effect (After 20 days of the pesticides being used). The results revealed that the insecticide residue in the seeds of all nine insecticide treatments was less than 0.01 mg kg^−1^ (Table 2).

Because Virtako appeared to be the most efficient candidate insecticide, its dynamic function was further monitored in 2021. We further analyzed the corncob damage rate every five days after the synthetic insecticide was used in the field (Figure 2c). It showed that, during the first 10 days, the control efficacy kept declining at a remarkably fast rate, especially for the sole Virtako application in sweet corn, which decreased by 31%. The control efficacy in the second 10 days decreased mildly, with decreasing rates varying from 1% to 7%. Mineral oil additive dramatically improved the persistence period of Virtako, in that the control efficiency was enhanced markedly, especially for sweet corn from 68% to 83% (Figure 2c). Furthermore, we tested the quantity of Chlorantraniliprole and thiamethoxam, the two major compounds in Virtako, to determine its life duration in sweet and waxy corn seeds after 3, 5, 6, 7, 10, 15, and 20 days of Virtako applied, and found that the quantity of Chlorantraniliprole in sweet and waxy corn was 0.00364 and 0.00215 mg kg^−1^ after three days, less than 0.01 mg kg^−1^, a generally accepted safe residual amount in vegetables [22]. However, degradation of Thiamethoxam, another systemic insecticide, to below its safety level (0.01 mg kg^−1^), took six and 16 days for sweet corn and waxy corn, respectively. In both corn varieties, the first six days were the fastest degradation period of Thiamethoxam, with a 69.19% residue decrease for sweet corn and 74.84% for waxy corn. After 20 days of Virtako use, the Thiamethoxam residues on sweet and waxy corn were 0.00175 and 0.00727 mg kg^−1^, respectively, whereas Chlorantraniliprole residues were under the limit of detection (0.01 µg/kg) in both corn varieties (Figure 2d).

### 3.4. Screening of High-Efficiency Biopesticide

To determine the optimal biogenic insecticides for corn pest control, we selected five bio-based pesticides for field testing in 2019: *B. bassiana*, Bt, Matrine, Veratrine, and Azadirachtin. *B. bassiana* + mineral oil treatment had the highest control efficiencies which were 88% and 80% for sweet and waxy corn, respectively, with 10% and 6% control efficiency improvement when compared to the *B. bassiana* treatment without mineral oil. Matrine also showed high efficiencies with 62% and 59%, respectively, in sweet and waxy corn. The control efficiencies for Bt + mineral oil were 57% and 55% on sweet and waxy corn, respectively, which were 4% and 7% higher than the Bt treatment without mineral oil. Azadirachtin and Veratridine appeared to be least effective against earworms of corn sweet (Figure 3a).

We then investigated the efficacy duration of *B. bassiana* in 2021, the most efficient biopesticide tested, with and without mineral oil on sweet and waxy corn, by measuring corn damage rate every five days after treatments. The data indicated varying efficiency changes. During the first ten days after the biopesticide application without mineral oil, there was a nearly 20% change in efficiency in the *B. bassiana* treatment on sweet and waxy corn. In the following ten days, the change ranged from 8% to 10%. We also observed that mineral oil could dramatically improve the effectiveness period of *B. bassiana*, and the control efficiency was enhanced, from 75% to 88% on sweet corn (Figure 3b).

### 3.5. Efficiency and Benefits Comparison of Different Insecticides among Corn Varieties

The efficiency of 14 different insecticides on different corn varieties was evaluated. The results showed that there was little difference between sweet corn and waxy corn for the three most effective insecticides, Virtako, Indoxacarb + Chlorfenapyr, and *B. bassiana.* Although the control efficiency of Beta-cypermethrin, Fenitrothion, and Phoxim differed, all three were ineffective for sweet corn ear pests (Figure 4).

We then calculated and compared the cost of insecticides with >50% control efficacy. The cost of Virtako plus mineral oil, the most efficient insecticide tested in this work, was the highest, which was $37.73 per hectare of cornfield. Indoxacarb + Chlorfenapyr had the best cost-benefit ratio among all synthetic insecticides with a cost of $11.67 per hectare and control efficiencies of 81% and 67% on sweet and waxy corn, respectively. The cost of Chlorantraniliprole per hectare of corn fields was $12.67 (Figure 4). Matrine was the lowest-cost biopesticide with a high control efficiency and its cost was $8.50 per hectare of corn. The *B. bassiana* + mineral oil cost $20.65, and *B. bassiana* without mineral oil cost $16.67. *Bacillus thuringiensis* plus oil cost $23.98 per hectare and its control efficiencies were 57% and 55% on sweet and waxy corn, respectively. Without mineral oil, *Bacillus thuringiensis* had a 4–7% lower control efficiency and a $3.98 lower cost per hectare of corn field (Table 1).

## 4. Discussion

We observed that the peak of insect pest density was highly correlated with the phenological stage of corn, which occurred during the silking stage of sweet corn when volatile production in corn is most appealing to lepidopterans. It has been reported that green leaf volatiles emitted by corn, including (Z) -3-Hexene-1-ol, (+)-Cyclosativene, β-Caryophyllene, could impact lepidopterans oviposition behavior on the host plant [23,24], by attracting unmated *Ostrinia nubilalis* adults and *Ostrinia furnacalis* larvae [25]. Environmental elements and habitat conditions also correlated to the population dynamics of lepidopteran pests during the corn ear stage. Appropriate temperature (22–30 °C) and humidity (RH more than 40%) are beneficial to pest survival, reproduction, and egg hatching [26]. Using insecticides at the right time is critical for accurate and effective pest management in sweet corn protection. Our data demonstrated that the control efficiency for insecticides applied in the silking period is significantly higher than that of the V12 stage insecticide spraying. Treatment at the V12 stage of corn in pest control experiments was chosen based on the facts that farmers control lepidopteran pests at this stage and that the large-scale occurrence of lepidopteran pests started at the V12 stage.

For Asian corn borer, cotton bollworm, and peach borer. Temperature and relative humidity are extensively used key climatic factors in lepidopteran pest outbreak prediction models, including cotton bollworm and corn borer [27]. Sexual attractants and frequency-vibration insect-killing lamps were used to monitor the insect in sweet corn fields, which can reduce systemic error because previous research has demonstrated that male *Conogethes punctiferalis* sensitivity to sexual attractants greatly changed after mating [28]. Thus, after the pollination of the sweet corn, the use of sexual attractants for pest collection might be no longer effective, but it could be complemented by light trapping. Our study may have some potential limitations. Firstly, we did not identify the insect species within corn ears, which is crucial for understanding pest populations and patterns. We plan to incorporate this aspect into our future experiments. Secondly, our insect monitoring and pesticide screening were limited to the period from May to June (the first stage of summer corn). We focused on this early period of time because we believe it provides a clearer insight into pest dynamics, with lighter overlap of generations and more valuable insights into pest behavior. However, pesticide effectiveness may slightly decrease in July and August, making pest control more challenging. We are currently preparing experiments to evaluate pesticide efficacy at different month.

Effective pest control was dependent on treatment time, insecticide efficiency, and the application technique [29]. We found that spraying Virtako + mineral oil on the silk during the silking period could achieve the highest control efficiency at approximately 87%. Virtako is an effective synthetic insecticide with Chlorantraniliprole and thiamethoxam as its main components. This occurred during the period of the first batch of summer corn when there was a limited overlap of pest generations and a relatively low pest population base. This partly explains the better control effect. Currently, it is widely used for controlling the pests of the orders of Lepidoptera, Coleoptera, Diptera, Hemiptera, and Orthoptera, such as rice borer *Chilo suppressalis*, sugarcane stem borer *Chilo sacchariphagus*, Asian corn borer, brown planthopper *Nilaparvata lugens* [30,31,32,33], and the control efficiency for lepidopteran pests generally ranged from 46% to 80%. We found that Thiamethoxam may have little to no activity on lepidopteran larvae because the main ingredients of Virtako are Thiamethoxam and Chlorantraniliprole, and the field control effect of Chlorantraniliprole alone is also good. but further experiments are still needed to confirm. In this study we showed that mineral oil considerably increased the efficacy of Virtako, supporting what has been previously reported [34], likely owing to some features of mineral oil such as changing the physical properties of insecticides, increasing their effectiveness, and extending their persistence. *B. bassiana* is a commonly used biopesticide to suppress the populations of lepidopteran larvae in corn, and spraying *B. bassiana* sporangial powder on cornstalk and the soil can kill overwintering lepidopteran larvae. The mineral oil could also improve the effectiveness of both microbial insecticides tested in this study, especially in sweet corn with mineral oil as a synergist. In fact, mineral oils have been shown to increase the control effect of *B. bassiana* by improving spore survival and adherence. For example, after adding emulsifiable mineral oil to *B. bassiana*, the mortality rate of *Listronotus maculicollis* larvae rose by more than 20% [35]. Interestingly, a study found that Dill oil from natural plants included dillapiole (5-allyl-6,7-dimethoxy-1, 3-benzodioxole) and displayed excellent insecticide synergistic activities [36]. Therefore, natural insecticide synergists should be exploited in the future to enhance the effectiveness of insecticides. In this study, we found that Azadirachtin and Veratridine were inefficient in killing corn pests, in spite that they both were toxic to some lepidopteran pests such as diamondback moth *Plutella xylostella* and cotton bollworm *H. armigera* [37]. The inefficiency of these two biopesticides in the field test may be due to their low toxicity. One contributing factor is the instability of larger molecular structures when exposed to suboptimal solvent conditions, such as variations in salt concentration and pH levels. These non-optimal solvent parameters can impact the stability and efficacy of the biopesticides. A study showed that a 500-fold dilution of the Azadirachtin solution, relative to its original concentration could be detoxicated on mulberry leaves and lose its silkworm toxicity within 30 min [38]. Hence, to test the toxicity of these compounds and their degradation on corn, and to protect them from degradation are the future directions that would help improve the efficacy of these pesticides.

In this study, we also examined the advantages of all insecticides based on their control efficiency and cost. According to our research, we recommend applying Indoxacarb + Chlorfenapyr, Virtako + mineral oil, or biopesticide *Beauveria bassiana* + oil on the surface of corn silks and ear axes within the first 5 days after corn silking to maximize effectiveness.

## Figures and Tables

**Figure 1 insects-14-00929-f001:**
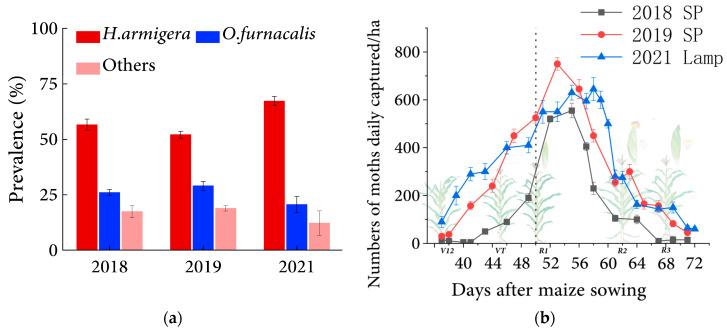
Relationship between insect pest adult’s occurrence and maize phenological stage. (**a**) The prevalence of maize pest adult at ear stage in Beijing in 3 years (means ± SE, 3 replicates per site. Data from 2018 to 2019 come from sex pheromone trapping marked as SP, and data from 2021 comes from the frequency-vibration insect killing lamp marked as Lamp. (**b**) The vertical dashed line denotes the beginning of maize silking and pollination, and it corresponds to the maximum moth population increasing rate. Error bars represent the Standard Error per site (n = 3). Missing values due to rain or other extreme weather conditions are not displayed in the report, full data are in Data1, Sheet 2. V12-R3 means different corn growth and development stages.

**Figure 2 insects-14-00929-f002:**
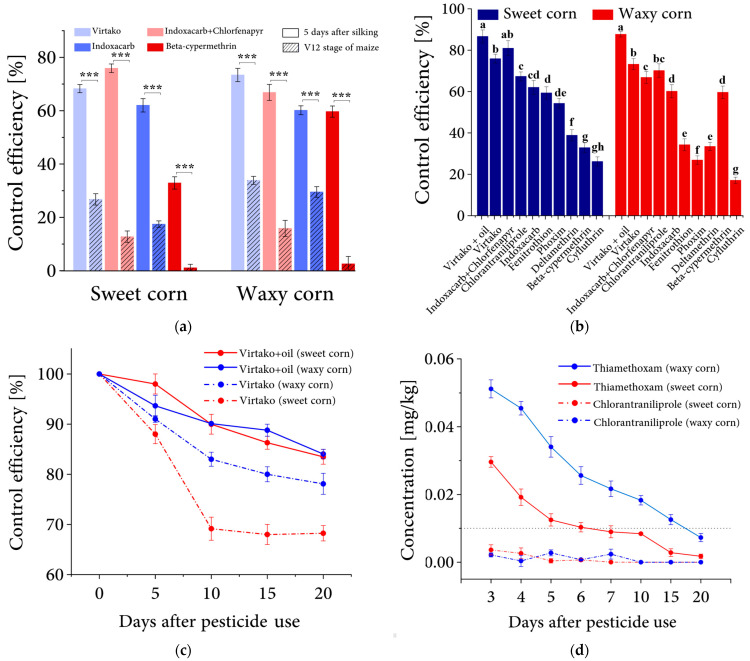
Control efficiency of chemical pesticides and pesticide residue on corn. (**a**) Control efficiencies in different phenological stages of sweet corn. Using insecticides in the R1 stage (five days after silking) and in the V12 stage of maize, respectively. Bars indicate the SE of control efficiency in three replicates for 150 corn plants each, and asterisks “***” indicate significant differences (*p* < 0.01) between two treatments by using independent-sample *t*-test. (**b**) The control efficiency of nine widely-used chemical pesticides in sweet and waxy corn. Pesticides were used five days after maize silking. Bars indicate the SE of control efficiency in three replicates. The letter marks on the bar indicate significance by using One-way ANOVA with Bonferroni–Holm post-test (*p* < 0.05). (**c**) The decreasing efficiency rule of Virtako in two treatments, One with mineral oil as an additive and another without. Spray insecticides once on the surface of the corncob five days after maize silking. Bars indicate the SE of control efficiency in three replicates. (**d**) Time-dependent reduction of the concentration of Virtako active compounds chlorantraniliprole and thiamethoxam on sweet and waxy corn grains. Bars indicate the SE of concentration, based upon three replicates. The dotted line indicates the EU pesticide residue limit (0.01 mg/kg) [21].

**Figure 3 insects-14-00929-f003:**
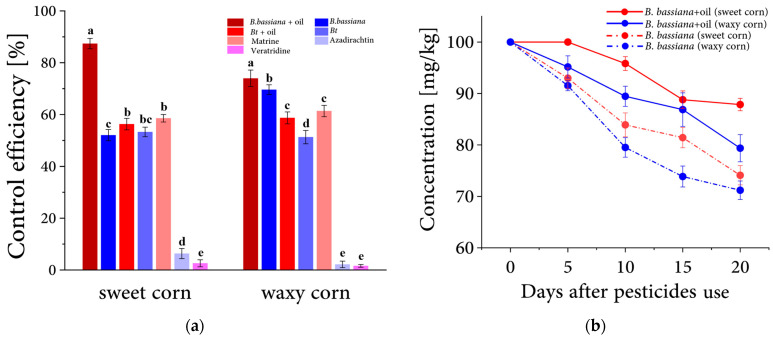
Control efficiency and the decreasing trend of biopesticides on sweet and waxy corn. (**a**) Control efficiency of five different biopesticides in sweet and waxy corn. Bars indicate the SE of control efficiency in three replicates. The letter marks on the bar indicate significance by using One-way ANOVA with LSD post hoc test (*p* < 0.05). (**b**) The efficiency-decreasing trend of the biopesticide *B. bassiana* in two treatments, one with mineral oil as an additive and another without. Bars indicate the SE of control efficiency (n = 3). Insecticides were applied once on the surface of the corncob at five days after maize silking.

**Figure 4 insects-14-00929-f004:**
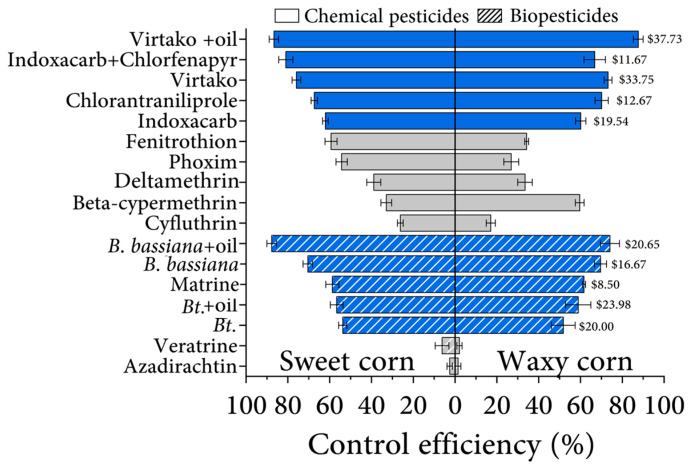
Different pesticide efficiency among maize varieties, and the pesticide cost. The first 9 treatments were chemical pesticide treatments and the last 7 were biopesticide treatments. The gray bar represents inefficient pesticides (control efficiency less than 50%), while the others are considered efficient. The cost of pesticides is on the right of each bar (US dollars hectare). Bars indicate the SE of control efficiency in three replicates Pesticides with a control efficiency of less than 50% were not calculated for cost.

**Table 1 insects-14-00929-t001:** All insecticides information.

	Name of Pesticide	Formulation	Content of Effective Constituent	Dosage (ha)	Cost ($/ha)
1	Deltamethrin	EC.	25 g L^−1^	600 mL	6.93
2	Fenitrothion	EC.	6%Fenvalerate + 14% Fenitrothion	900 mL	6.40
3	Phoxim	EC.	40%	750 mL	4.43
4	Chlorantraniliprole	EC.	200 g L^−1^	150 mL	12.67
5	Virtako	WDG	40% (20% Chlorantraniliprole + 20% thiamethoxam)	120 g	33.75
6	Indoxacarb + Chlorfenapyr	EC.	10% (5%Indoxacarb + 5%Chlorfenapyr)	600 mL	11.67
7	Indoxacarb	EC.	150 g L^−1^	375 mL	19.54
8	Beta-cypermethrin	AR.	4.5%	750 mL	3.70
9	Cyfluthrin	AR.	5%	300 mL	5.12
10	*Beauveria bassiana*	Pulv	5 × 10^9^ spores g^−1^	6000 g	16.67
11	*Bacillus thuringiensis* (Bt)	Pulv	50,000 IU mg^−1^	900 g	20
12	Azadirachtin	EC.	3%	2250 mL	73.20
13	Matrine	Water aqua	1.3%	600 mL	8.50
14	Veratrine	SC.	0.5%	2250 mL	3.65
15	Mineral oil	Additives	99 %	3000 mL	3.85

EC. = Emulsifiable concentrate. WDG = Water Dispersible Granules. AR = Aqua Resigen. Pulv = Pulverize (powder formulation). SC.= soluble concentrate. Water aqua = Water-based formulation.

**Table 2 insects-14-00929-t002:** Synthetic pesticide content in corn grains 20 days after insecticides had been applied (µg kg^−1^; mean ± SE). LoD (0.01 µg/kg).

	Treatments	Jingketian (Sweet Corn)	Jingkenuo (Waxy Corn)
1	Beta-cypermethrin	<LoD	<LoD
2	Indoxacarb	0.20 ± 0.02	0.01 ± 0.01
3	Indoxacarb + Chlorfenapyr	0.37 ± 0.04	0.08 ± 0.01
4	Chlorantraniliprole	0.39 ± 0.02	0.60 ± 0.05
5	Cyfluthrin	0.05 ± 0.01	<LoD
6	Virtako	0.25 ± 0.03	5.22 ± 0.23
7	Deltamethrin	<LoD	1.20 ± 0.09
8	Fenitrothion	<LoD	<LoD
9	Phoxim	0.04 ± 0.01	0.07 ± 0.02
10	Control	<LoD	<LoD

## Data Availability

Data is contained within the article or Appendix A.

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
