# Peer review of "The Efficiency of Pest Control Options against Two Major Sweet Corn Ear Pests in China"

_insects, 2023, doi:10.3390/insects14120929_

Round 1

Reviewer 1 Report

Comments and Suggestions for Authors

Overall a lot of work was conducted and there is definitely a merit in publishing the obtained results. Many valuable data were obtained, but the main concerns are with the presentation of the results. However, the paper requires extensive editing, grammar, sentence wording, paragraph structure, and organizing the body of each section. I did not correct the issues throughout the all paper because there are too many changes that need to be made. 

The introduction is too long, and excessive irrelevant information is included meanwhile the important citation is either not included or poorly structured. It should be shortened and structured with the hypothesis and objectives clearly stated. The introduction will benefit if only a relevant focused analysis of previous research is conducted.

The materials and Methods section needs a major improvement. As it is written now it is impossible to repeat/replicate this experiment.  Authors omit important details and it is hard to judge the quality of the design and research. for instance how the treatments were delivered. If sprayed, what type of sprayer is used with how much output per plant? 

The description of statistical analysis has to be significantly improved. The variables of interest have to be mentioned, what factors were considered? Were timing and addition of adjuvant considered as factors?

How was the percent of the control calculated? What formula was used?

The result section includes a lot of explanations and speculations which have to be in the discussion. This makes understanding of the results difficult and unnecessarily laborious. I would suggest to present data clearly with more details but no explanation in this section,

I would look at the discussion section again after the other parts are fixed.

Please see some suggestins and comments in the attached document.

Comments on the Quality of English Language

Hard to read and understand. Needs significant improvement.

Author Response

Response to Reviewer 1 Comments

I want to express my sincere gratitude for the meticulous feedback you provided. Here are the primary revisions:

Point 1: The introduction is too long, and excessive irrelevant information is included meanwhile the important citation is either not included or poorly structured. It should be shortened and structured with the hypothesis and objectives clearly stated. The introduction will benefit if only a relevant focused analysis of previous research is conducted.

Response 1: Thanks for the suggestion. We have reduced the length of the introduction, added hypotheses and objectives. The entire manuscript has undergone a comprehensive revision, involving reorganization, editing, and trimming across the introduction, conclusion, and discussion sections.

Point 2: The materials and Methods section needs a major improvement. As it is written now it is impossible to repeat/replicate this experiment.  Authors omit important details and it is hard to judge the quality of the design and research. for instance how the treatments were delivered. If sprayed, what type of sprayer is used with how much output per plant? 

Response 2: The "Methods" section has been substantially improved, with a strong emphasis on providing detailed information to elucidate the experimental procedures and make replication feasible. All the details you mentioned have been filled in.

Point 3: The description of statistical analysis has to be significantly improved. The variables of interest have to be mentioned, what factors were considered? Were timing and addition of adjuvant considered as factors?

Response 3: Thank you for your valuable advice about statistical analysis. We have added the "variables of interest" and "factors" as per your request. In the first experiment, the factors include "treatment time" (V12 spraying and 5 days after spraying), and the "control efficiency" serves as the variable of interest. In the second experiment, the factors include "pesticide type," and the "control efficiency" serves as the variable of interest. It's important to note that the addition of an adjuvant was only used in two pesticide treatments (B. bassiana and Virtako) and was not treated as a separate factor throughout the entire experiment. At the same time, we also corrected our statistical methods. Statistical methods have been upgraded from the previous "one-way ANOVA with LSD post hoc test" to "one-way ANOVA with the Bonferroni-Holm post hoc test" to prevent alpha error inflation. The revised statistical results are now clearly presented in Figure 2b.

Point 4: How was the percent of the control calculated? What formula was used?

Response 4: We’ve added formulas of Control efficiency and Corn damage rate in methods.

Point 5: The result section includes a lot of explanations and speculations which have to be in the discussion. This makes understanding of the results difficult and unnecessarily laborious. I would suggest to present data clearly with more details but no explanation in this section.

Response 5: Thanks for your advice. We have revised the entire results section based on your suggestions.

Point 6: if this method used separately only for one year it is probably can be omitted. It does not add any information.

Response 6: Thanks for the advice. I have added an explanation in the results section regarding why frequency-vibration insect-killing lamps were used in the third year. We want to avoid the error caused by a single method experiment.

Thank you again for your invaluable feedback, which motivated us to think about our work, and it has significantly improved the quality and of our work.

Reviewer 2 Report

Comments and Suggestions for Authors

Comments on the Quality of English Language

See the attached document under comments and suggestions for authors section

Author Response

Response to Reviewer 2 Comments

I want to express my sincere gratitude for the meticulous feedback you provided. Here are the primary revisions:

Point 1: Do you mean to say that the silks were coated in insecticide?

Response 1: Thanks for the comments. Yes, we applied the pesticide during the silk stage, targeting only the silks and ear surface. We believe that protecting the corn ears, rather than the entire plant, yields higher benefits.

Point 2: Describe how the corn was evaluated for damage. Were worm species identified and tallied? This is extremely important as Asian corn borer may infest pre-silking corn plants and then tunnel into the ear whereas H. armigera is much less likely to do so. Were larval sizes estimated? When ears were evaluated every 5 days, were 150 plants evaluated or does the 150 represent a total # evaluated over the course of the experiment or just at the final harvest? What varieties of corn were planted? When? What was the row spacing, plant population? Are Jingketian and Jingkenuo variety names or other names to describe sweet corn and waxy corn?

Response 2: the "Methods" section has been substantially improved, and both damage rate and control efficiency have been included. We did not identify the worm species in this experiment, which is a limitation of our study. We have added the "worm species identification" to the discussion section. On the other hand, we have two preliminary ways of determining whether it's the cotton bollworm or corn borer larvae when assessing the ear damage rate: first, by examining their excrement, as cotton bollworm larvae tend to produce more excrement; second, by inspecting the boreholes on the surface of the corn ear husks, as these are generally caused by corn borers, while cotton bollworms tend to invade more from the silk part. We have also added the specific method for evaluating ears to the "Methods" section. In short, for each pesticide treatment, we had three replicates, each consisting of 150 corn plants, totaling 450 corn plants. We repeated the assessment every 5 days to monitor the extent of ear damage rate increase. All the other details you mentioned have been incorporated.

Point 3:  As measured by what type of trap? Delta traps do not catch other Helicoverpa well, so your moth flight might be even greater.

Response 3: Thank you for providing the information. Yes, we also considered that the pheromone trap method might have its limitations, so we switched to light trapping in the third year to reduce potential systematic errors.

Point 4: Line 234: change ‘validity period’ to residual activity If it is warm enough for sweet corn to be growing and silking, it is warm enough for moths to be flying. The sentences between lines 230-233 are obvious. Worms in harvested sweet corn are not acceptable. Thus they must be controlled before they get into the ear. These sentences do not add anything.

Response 4: We have made changes to delete the sentences without information.

Point 5: Figure 1. ‘Others’ does not mean anything unless they also get into the ear, but there are no ear worm species identifications. spelling errors on Y axis of b. Break out figure b by moth species. This chart does not mean much by itself without species indicated.

Response 5: Thank you for your feedback. Indeed, conducting ear worm species identifications would enhance the credibility of the results. We did not perform ear worm species identifications due to several considerations. Firstly, it is an extremely labor-intensive task, and secondly, the larval morphology can be challenging to identify, especially for early-stage larvae like the Grapholita molesta and Ostrinia furnacalis. Additionally, there are instances where, during the final ear assessment, we observed corn damage without finding any larvae inside. In our future work, we will consider incorporating data from ear worm species identifications using suitable methods. Nevertheless, we believe that the figures have its meaning, as the category "others" consists of the adult stages of pests that directly impact corn, making them relevant. Thank you once again for your valuable insights.

Point 6:  Line 251: were the insecciticdes applied 2x per plot (V12 and R1+5 days) or applied one time per plot treatment, two plot treatments being timing?

Response 6: We have thoroughly revised the "Methods" section. Each treatment involved a single pesticide application. V12 treatment entailed applying the pesticide during the V12 stage, while "5 days after silking" treatment involved a single pesticide application specifically 5 days after silking.

Point 7:  Line 258-259: How can Virtako rank second if its % clean ears is greater than Indoxacarb+Chlorfenapyr?

Response 7: The rank was based on the magnitude of increase control efficiency. I apologize for any misunderstanding, and I have now revised this statement.

Point 8: Section 3.3 Remind the reader what year the data presented in this section were taken. Or are multiple years combined into a single analysis?

Response 8: We added the detailed time of each experiment into methods.

Point 9: Is evaluating pesticide efficacy at multiple time intervals even necessary? Once the worm is in the ear, no amount of treating is going to kill that worm or result in a clean ear once it has been damaged.

Response 9: This is a great question, and we also believe that there wasn't a significant need for multiple assessments. Most of our data only presents the final control efficiency data at the harvest stage since the earlier control efficiency data did not show a strong pattern in most of the cases.

Point 10: It may be useful to have multiple plantings, with the silking period separated by several weeks to assess efficacy. In parts of the U.S., Helicoverpa zea populations wax and wane, and their response to insecticide application also changes throughout the course of the season. Peak populations in August and September are difficult to adequately control even with 5 or more insecticide applications.

Response 10: Thank you for providing valuable insights. Multiple plantings are also common in Beijing for sweet corn. We considered an important factor in our experimental timing: we chose May, which is essentially the first batch of sweet corn planting. At this time, there are only first generations of moths in the field, and the problem of overlapping generations is not severe. By August, as you mentioned, the pest population increases, and their migratory behavior becomes more pronounced. It is indeed challenging to detect their patterns and implement effective control measures at that stage. The best approach in late August is either to sever the source of moths or to maintain a consistently high level of pesticide efficiency on the corn ears through frequent applications. Our data on moth density in August showed that the maximum moth density was three times that of the data from this study.

Point 11: Line 291: Chloranbenzoamide was not mentioned in the text. Check to make sure this wasn’t meant to be chlorantraniliprole.

Response 11: Thank you for your careful review. I made a mistake, which I have corrected. It should be Chlorantraniliprole.

Point 12: In the discussion, you might want to point out that Chlorantraniliprole and Virtako are essentially the same type of insecticide. The Thiamethoxam component has little to no activity on Lep larvae.

Response 12: Thank you for the reminder. Yes, I believe this is an excellent point to initiate the discussion, and we have already included it in the discussion section.

Point 13: Line 297-300: Was any phytotoxic response observed due to the mineral oil?

Response 13: We did not observe any significant signs of plant toxicity, and your reminder has prompted us to closely monitor the condition of the plants that were treated with mineral oil in the future.

Point 14: Line 337: What brand and strain of B. bassiana was used? Same question for Table 1: please give brand and manufacturers. Please give product rate/hectare.

Response 14: B. bassiana is of the "Pulverize" type (powder formulation) with a concentration of 5x10^9 spores per gram. It is sourced from the Chinese Academy of Agricultural Sciences, Plant Protection Institute, and is priced at $16.67 per hectare, as shown in Figure 4.

Point 14: Line 369: In the discussion, based on the cost information presented, what would be your recommended product and product use pattern to achieve the greatest economic benefit?

Response 14: We have made recommendations in the discussion section based on our results.

Point 15: Line 388-393: These sentences should be excluded.

Response 1: Have deleted

Point 15: Line 432: what is a 500-fold solution?

Response 15: I have corrected the expression of the sentence.

Point 16: In the United States when other Helicoverpa populations peak in the 20+ per night range in wire mesh pheromone traps, which are the most sensitive pheromone trap design, 5 or more insecticide applications are necessary to keep corn worm free. If those applications are not timed appropriately (no more than 3 days after first silk), the field will suffer heavy damage. I have a hard time believing that a single application 5 days after first silk can be so effective. That any control was observed with V12 applications is suspect unless the main target was Asian corn borer, but this is hard to know because species are not reported. I have a hard timing believing that B. basiana can protect ears better than chlorantraniliprole. What is considered a control efficiency – lack of live larvae or a non-worm fed upon ear?

Response 16: Thank you for providing such valuable information. You are absolutely correct. I would like to ask if this control strategy is implemented during warmer temperatures, such as late July or August? In China, around August, all pesticide applications need to be done at least two days in advance. Farmers often start spraying at the silking stage because this is when there's significant overlap in moth generations, a high pest population, and strong moth migration capabilities. Considering the time needed for spraying, the action threshold for pest infestations must be lowered to proactively address the complex pest situation. It seems that targeting three days after silking aligns well with this timeframe. During this period, as you mentioned, using V12-stage spraying is unlikely to be effective. We also avoided conducting the experiment in July and August.

Point 17: There are inconsistent punctuations, periods, and capitalizations with internal citations throughout the manuscript. In some places periods are absent after citations which end the sentence, and in other places they are before the citation. There are missing spaces, extra spaces, and names that need to be capitalized throughout.

Response 17: We have carefully reviewed and corrected the detail errors from beginning to end.

Thank you again for your invaluable feedback, which motivated us to think about our work, and it has significantly improved the quality and of our work.

Reviewer 3 Report

Comments and Suggestions for Authors

I have problems to accept this paper:

- the title suggests inappropriate generalizations, claiming 

*applicability to all ear pests while only 2 major pest species are eplored in the observations and

* global applicability while all experiments were only conducted in China (This is also reflected by the literature, 28 of 48 sources originating from Chines authors). It may really be doubted that a series of 3 experiments over 3 years may suffice as a basis to be generalized to worldwide validity

- The trappíng method has been changed throughout the experiment, from pheromone traps 2018-19 to insect killing lamps 2020. There is no good reason given, and no detailed discussion to convince the readers why such would not distort the results.  Figure 1 shows the massive differences, and there is no compelling explanation why this should not matter.

- Statistical methodology is not up-to-date, still using LSD for comparison between groups. There is multiple testing in place, and it would be preferrable to apply Sidak*s method or Bonfereoni-Holm to avoid alpha error inflation. Instead of standard error, 95% confidence intervals should be applied. There is no mention of missing data, nor, how to deal with them.

There are also minor faults in the manuscript which cast doubts on the scientific quality and reliability of this paper:

- the paper mixes dimensions (acres and square meters; 1/6 acre is equivalent to 674.5 m², and not 667m²

- There is no LoD (level of determination) and no LoQ (level of quantification) reported for the chemicval analyses

- perecentage is given to a precision of 2 decimal digit fraction, which suggest a level of precision that is not really there; rounding to full percentage would be appropriate

- error bars are shown in the graphs, but as SE=standard error; this avoids that reades have a full benfit, whicht tzhey would have from 95% confidence intervals

- there is no menmtion whether the methods have been validated (as being requeste3d by GLP (Good Laboratory Practice)

-  the dimension of pesticide residuals is expressed (table 2) in mg/kg.
Such gives the impression of very low levels, and it is hard to read with many zeroes after the decimal point. Such should be expressed alt least as µg/kg ( e.g. 0.00037 mg/kg = 0.37µg/kg. )

- Table 2 also specifies zero measurements  as 0.00000, which sould better be expressed as "< LoD" and LoD be reported, or as "<0.01µg/kg"

- Table 1 refers dosage and cost to square meters obviously, but the last two columnjs are named as "Dosage (667 ²)" and "Cost (y 667-²)" which should read correctly asd "Dosage (per m²)" and "Cost (per m²)"

I SUGGEST
(1) TO CHANGE THE TITLE, ADDING RESTRICTIONS TO "TWO MAJOR CORN PESTS IN CHINA".

(2) TO CAREFULLY REVISE THE TEXT; REMOVE IMPRECISION AND INCONSISTENCIES.

(3) TO DISCUSS THE WEAKNESSES (LIKE CHANGE OF TRAPPING METHOD)

Comments on the Quality of English Language

I did not check such features to avoid belated feedback

Author Response

Response to Reviewer 3 Comments

I want to express my sincere gratitude for the meticulous feedback you provided. Here are the primary revisions:

Point 1: - the title suggests inappropriate generalizations, claiming.  *Applicability to all ear pests while only 2 major pest species are eplored in the observations and

Response 1: Thank you for your suggestion, and we have revised the title accordingly. The title has been refined to "The Efficiency of Selective Ear Pest Control Methods in Sweet Corn in China," narrowing the scope.

Point 2: * global applicability while all experiments were only conducted in China (This is also reflected by the literature, 28 of 48 sources originating from Chinese authors). It may really be doubted that a series of 3 experiments over 3 years may suffice as a basis to be generalized to worldwide validity

Response 2: Thank you for your suggestion. We have rephrased the language in the text related to global applicability, narrowing the scope. The reference citations have also been reorganized accordingly.

Point 3: The trappíng method has been changed throughout the experiment, from pheromone traps 2018-19 to insect-killing lamps 2020. There is no good reason given, and no detailed discussion to convince the readers why such would not distort the results.  Figure 1 shows the massive differences, and there is no compelling explanation why this should not matter.

Response 3: We have made the necessary changes and included the explanation in the text. We considered that pheromone trap methods might have their limitations, which is why we switched to light trapping in the third year to reduce potential systematic errors.

Point 4: - Statistical methodology is not up-to-date, still using LSD for comparison between groups. There is multiple testing in place, and it would be preferable to apply Sidak*s method or Bonfereoni-Holm to avoid alpha error inflation. Instead of standard error, 95% confidence intervals should be applied. There is no mention of missing data, nor, how to deal with them.

Response 4: the "Methods" section has been substantially improved, with a strong emphasis on providing detailed information to elucidate the experimental procedures and make replication feasible. We have introduced formulas for calculating corn damage rates and control efficiency. The field survey and statistical methods have been expanded and refined. Statistical methods have been upgraded from the previous "one-way ANOVA with LSD post hoc test" to "one-way ANOVA with the Bonferroni-Holm post hoc test" to prevent alpha error inflation. The revised statistical results are now clearly presented in Figure 2b.

Point 5: There are also minor faults in the manuscript which cast doubts on the scientific quality and reliability of this paper:

Response 5: The entire manuscript has undergone a comprehensive overhaul, involving reorganization, editing, and streamlining across the introduction, conclusion, and discussion sections. The errors and unclear statements you mentioned throughout the paper have been addressed and corrected. In our study, we place special emphasis on ensuring the effectiveness of our experimental methods and compliance with the Good Laboratory Practice (GLP) standards.

Point 6: I SUGGEST (1) TO CHANGE THE TITLE, ADDING RESTRICTIONS TO "TWO MAJOR CORN PESTS IN CHINA". (2) TO CAREFULLY REVISE THE TEXT; REMOVE IMPRECISION AND INCONSISTENCIES. (3) TO DISCUSS THE WEAKNESSES (LIKE CHANGE OF TRAPPING METHOD)

Response 6: Thank you for your suggestion. We have revised the title, and we have thoroughly and meticulously revised the entire manuscript, reorganizing it to eliminate errors and inaccuracies. We have also included the weaknesses of the article in the discussion section.

Thank you again for your invaluable feedback, which motivated us to think about our work, and it has significantly improved the quality of our work.

Reviewer 4 Report

Comments and Suggestions for Authors

Li et al. present the results of field studies on the effects of synthetic and biological insecticides on lepidopteran pests of corn ears in Beijing, China.
The manuscript is not very clear and requires major improvement.
Most importantly, the authors need to detail how they measured and calculated crop damage and insecticide efficiency, as these calculations are not explained, so it is impossible to understand the meaning or significance of the results.
I have written numbered points and numerous suggestions on a scanned copy of the manuscript.
Numbered points (see scanned manuscript).
1.    You mean that larvae inhabit the ears and are protected by the husks? (husks, not hucks)
2.    Define monitoring period. Specify that the corn fields were located in Beijing, China.
3.    The international units for area are hectares. Please convert all area measurements to hectares (ha) in the manuscript.  I calculate that this is equivalent to 549 – 757 moths/ha, correct?
4.    You mean that this many moths were captured per day? Unclear.
5.    This level of pesticide reside is safe for food consumption according to who? Chinese government?
6.    Is this the value of corn output in China? Specify.
7.    This example is not relevant. Delete text.
8.    Husks, not hucks.
9.    This text on timing is much too long. Please summarize.
10.    City, Country of origin of traps?
11.    Use hectares or m2 as units of area (one sixth of an acre is 674 m2)
12.    Specify the period during which pheromone traps were used.
13a. Specify duration of light trapping (how many days or weeks each year?)
13b. Move Table 1 to here. Are these all commercial formulations of the chemicals that you mentioned? Were they all products produced in China?
14.    Specify – a replicate comprised a block of 667 m2 correct?
15.    If you add 200 ml of oil to water you get two phases. You need to specify that you used EMULSIFIABLE oil with detergents or surfactants.
16. How did you measure and calculate corn damage?
17. How did you measure and calculate control efficiency?
18. How many samples were analyzed and how was this study replicated?
19. How did you check for equal variances in the t-tests? How did you check for normality and homoscedasticity in the data used for ANOVA?
20. Where were the weather stations located (how far away from your experimental fields?) for the meteorological data?
21. y-axis label should be “Numbers of moths captured/ha” (change to hectares)
22. What does the dotted line indicate in Fig 2d?
23. It is not possible to understand the results as there is no explanation of what “control efficiency” means.
Please indicate pest infestation levels (e.g. how many corn ears had larvae present).
24. You mean “20 days after insecticides had been applied”?
25. These residue values do not agree with the values shown in Table 2.
26. What does “control effect” mean?
27. Rule? Please explain or reword.
28. What does “damage rate” mean?
29. dropping rates? Meaning?
30. A 13% increase is not a “drastic” increase.
31. You should mention the names of the corn varieties in the Methods section.
32. Change to dollars per hectare.
33. What do the other colored bars indicate in Fig 4?
34. What does Pulv/Pulverize formulation mean in Table 11? Are these powders? Or liquids or some other type of formulation?
35. Phoxim = 0.4 (Table 1) but 0.4 of what? g/L? mg/L? 0.4% (wt/vol)?  Please explain.  
36. The same applies to the other insecticides shown in Table 1. Indicate units of concentration of the active ingredients.
37.  Table 1. Please convert costs to US dollars/hectare, as you have shown dollars in Figure 4.
38. The main issue with botanical insecticides is their low potency/toxicity compared to synthetic compounds.
39. 500-fold azadirachtin? 500-fold of what? Please explain the units of concentration.
40. What about the impact of these compounds on insect natural enemies? Does China have a policy of preferentially using insecticides that conserve predators and parasitoids?
41. The references have many errors and missing information.

Comments on the Quality of English Language

Needs editing.

Author Response

Response to Reviewer 4 Comments

I want to express my sincere gratitude for the meticulous feedback you provided. Here are the primary revisions:

Point 1: The manuscript is not very clear and requires major improvement. Most importantly, the authors need to detail how they measured and calculated crop damage and insecticide efficiency, as these calculations are not explained, so it is impossible to understand the meaning or significance of the results.

Response 1: The entire manuscript has undergone a comprehensive overhaul, involving reorganization, editing, and streamlining across the introduction, conclusion, and discussion sections. The "Methods" section has been substantially improved, with a strong emphasis on providing detailed information to elucidate the experimental procedures and make replication feasible. We have introduced formulas for calculating corn damage rates and control efficiency. The field survey and statistical methods have been expanded and refined.

Point 2: You mean that larvae inhabit the ears and are protected by the husks? (husks, not hucks)

Response 2: Response 1: Yes, once pests enter the corn ears, conventional pesticide spraying is generally ineffective.

Point 3: Define monitoring period. Specify that the corn fields were located in Beijing, China.

Response 3: We have included that information.

Point 3: The international units for area are hectares. Please convert all area measurements to hectares (ha) in the manuscript. 

Response 3: Thank you for your advice. We have standardized the units to hectares throughout the entire manuscript, as per your suggestion.

Point 4: You mean that this many moths were captured per day? Unclear.

Response 4: In the light trapping in 2021, yes, the data was collected daily.

Point 5: This level of pesticide reside is safe for food consumption according to who? Chinese government?

Response 5: This includes the pesticide residue limits recognized in most regions, including China and the European Union. Supporting data has been added.

Point 6: This text on timing is much too long. Please summarize.
Response 6: The summarization has been redone.

Point 7: 3a. Specify duration of light trapping (how many days or weeks each year?) 13b. Move Table 1 to here. Are these all commercial formulations of the chemicals that you mentioned? Were they all products produced in China?

Response 7: The specific times for light trapping have been added to the "Methods" section. Most of the pesticides used are produced in China and are among the most widely used by farmers in the Beijing area.

Point 8: 15. If you add 200 ml of oil to water you get two phases. You need to specify that you used EMULSIFIABLE oil with detergents or surfactants.

Response 8: I have included this information as per your suggestion.

Point 9: 16. How did you measure and calculate corn damage? How did you measure and calculate control efficiency?

Response 9: I have added the corresponding formulas to the "Methods" section of the document.

Point 10: 18. How many samples were analyzed and how was this study replicated?

Response 10: It has been added to the "Methods," specifically, with three corn ears per treatment as replicates.

Point 11: 19. How did you check for equal variances in the t-tests? How did you check for normality and homoscedasticity in the data used for ANOVA?

Response 11: The statistical methods have been revised, and the information regarding the normality and homoscedasticity of the data has been included.

Point 12: 20. Where were the weather stations located (how far away from your experimental fields?) for the meteorological data?
Response 12: The specific information has been added to the document.

Point 13: 23. It is not possible to understand the results as there is no explanation of what “control efficiency” means. Please indicate pest infestation levels (e.g. how many corn ears had larvae present).
Response 13: I have added all the missing information to the "Methods" section.

Point 14: 25. These residue values do not agree with the values shown in Table 2.
Response 14: Thank you for your keen observation. These residue values are in Figure 2d, not in the content of Table 2. I have also improved the wording to make it less ambiguous.

Point 15: 33. What do the other colored bars indicate in Fig 4?

Response 15: Pesticides with control efficiency below 50%, representing a low level of efficacy, are not recommended.

Point 16: 34. What does Pulv/Pulverize formulation mean in Table 11? Are these powders? Or liquids or some other type of formulation? Phoxim = 0.4 (Table 1) but 0.4 of what? g/L? mg/L? 0.4% (wt/vol)?  Please explain.  The same applies to the other insecticides shown in Table 1. Indicate units of concentration of the active ingredients.
Response 16: I have added the information.

Point 17: 39. 500-fold azadirachtin? 500-fold of what? Please explain the units of concentration.

Response 17: I have made modifications to the original text to avoid ambiguity.

Point 18: 40. What about the impact of these compounds on insect natural enemies? Does China have a policy of preferentially using insecticides that conserve predators and parasitoids?

Response 18: These compounds also have strong toxicity to natural enemies. Currently, there is a policy in China that prioritizes the use of natural enemies and biopesticides, but full implementation may require some time.

Thank you again for your invaluable feedback, which motivated us to think about our work, and it has significantly improved the quality of our work.

Round 2

Reviewer 3 Report

Comments and Suggestions for Authors

I foun d some minor flaws:

General remark: dimensions should be enclosed in "[]" brackets. Example [%]

line 100 should contain "84" instead of "83.70"

Figure 2 legend to graph 2d currently explains: "(d) Pharmacodynamic decrease regularity of Virtako on sweet and waxy corn grains. Chlorantraniliprole and thiamethoxam are major constituents of Virtako". The expression "decrease regularity" is a very uncommon term in pharmacokinetic context. Moreover "pharmacodynamic" is the wrong term, as the graph obviously does not dispaly the weaning efficacy of the active compounds; as the graph displays the "content", this is a matter of pharmacokinetics. Therefore the y axis legend of graph 2d should be changed from "Content (mg/kg)" to "Concentration [mg/kg]".

Proposal for the figure 2d legend: "(d) Time-dependent reduction of the concentration ov Virtako active compounds Chlorantraniliprole and thiamethoxam on sweet and waxy corn grains.".

line 315 should be reformulated by: "Bars indicate the SE of concentrationm, based upon  three replicates."

line 318: better expression than "select" would be "identify"

line 378: instead of "efficacy" use "efficiency"

line 384 instead of "duration period" use "effectiveness period"

line 681-2: separate as "bio-pesticide"

Comments on the Quality of English Language

overall fine, just some expressions to be corrected

Author Response

 Response to Reviewer Comments

Point 1: General remark: dimensions should be enclosed in "[]" brackets. Example [%]

Response 1: Thank you for your advice. I have revised the whole manuscript.

Point 2: line 100 should contain "84" instead of "83.70"

Response 2: Thanks for your careful check, we have revised it.

Point 3: Figure 2 legend to graph 2d currently explains: "(d) Pharmacodynamic decrease regularity of Virtako on sweet and waxy corn grains. Chlorantraniliprole and thiamethoxam are major constituents of Virtako". The expression "decrease regularity" is a very uncommon term in pharmacokinetic context. Moreover "pharmacodynamic" is the wrong term, as the graph obviously does not dispaly the weaning efficacy of the active compounds; as the graph displays the "content", this is a matter of pharmacokinetics. Therefore the y axis legend of graph 2d should be changed from "Content (mg/kg)" to "Concentration [mg/kg]".

Proposal for the figure 2d legend: "(d) Time-dependent reduction of the concentration of Virtako active compounds Chlorantraniliprole and thiamethoxam on sweet and waxy corn grains.".

line 315 should be reformulated by: "Bars indicate the SE of concentration, based upon three replicates."

Response 3: Thank you for providing us with the knowledge of pharmacokinetics. We also feel that our words are not good, and we have modified it according to your suggestion.

Point 4: line 318: better expression than "select" would be "identify"

Response 4: Thanks for your suggestion, it has been modified

Point 5: line 378: instead of "efficacy" use "efficiency"

Response 5: It has been modified.

Point 6: line 384 instead of "duration period" use "effectiveness period". line 681-2: separate as "bio-pesticide"

Response 6: All modified.

Thank you for revising our manuscript so carefully. We appreciate and cherish it very much!

Reviewer 4 Report

Comments and Suggestions for Authors

The authors have addressed many of my concerns, but I still find that the manuscript contains errors and inconsistencies that need to be addressed.

The title is grammatically incorrect. Should be changed to "The efficiency of pest control options against two major sweet corn ear pests in China"

Line 50. Order is "Lepidoptera" (members of this order are "lepidopterans")

Line 90. Change "form of reagents " to "compounds".

Line 94. Change biological to "biologicals"

L96. Trichogramma (italics)

L103. "most wasps are specialists attacking a single host insect species" – what is your evidence for this? In my experience most parasitoids attack several host species.

L107. You say that B. bassiana has low efficacy for ear pest control, but in the summary you recommend the use of this pathogen. Please clarify.

L126. Planting density "density of 30cm 60cm" What does this mean? Usually densities are written as X thousand plants/ha

Throughout manuscript. Change "0.0667 ha" to 667 m2.

L154. Change to 333 m2.

L162 (Table 1) What type of B. thuringiensis was used? var. kurstaki?

L169. Delete "per 0.0667 hectare".

Table 1. What is the dosage and cost unit "he"?? Should this be hectare (ha)?

Line 328 and Table 2 and Fig 4. (+data1 file, sheet 9) Change Decamethrin to Deltamethrin.

Table 2. What does LoD mean?

Fig 3B – how can B. bassiana have a control efficiency of 100% at 0 days post-application? This biological insecticide takes several days to begin killing insects.

Fig 3A. Why doesn't the control efficiency of B. bassiana in (A) 78-88% match the control efficiency in (B)? Even if you average the points in B you get higher values than shown in A.

L454. "We found that thiamethoxam, may have little to no activity on lepidopteran larvae" – where is your evidence for this assertion? You did not test this.

L477. What do you mean by "buffer conditions"? Please reword or clarify in text.

L484. "The findings revealed that most biopesticides are more costly than synthetic insecticides." According to Fig 4. this is not correct. The average cost of the five best chemical insecticides was US$22.9/ha according to my calculations, whereas the five best biopesticides had a cost of US17.9/ha. 

Supplemental material - data1.xls file, sheet 9 – change $/acre values to $/hectare.

Comments on the Quality of English Language

Needs editing.

Author Response

Response to Reviewer Comments

Point 1: The title is grammatically incorrect. Should be changed to "The efficiency of pest control options against two major sweet corn ear pests in China"

Response 1: Thank you for your suggestion. We have made the corrections, and we believe the revised title is now very clear.

Point 2: Line 50. Order is "Lepidoptera" (members of this order are "lepidopterans"). Line 90. Change "form of reagents " to "compounds". Line 94. Change biological to "biologicals". L96. Trichogramma (italics)

Response 2: Thank you, all of the above have been revised.

Point 3: L103. "most wasps are specialists attacking a single host insect species" – what is your evidence for this? In my experience most parasitoids attack several host species.

Response 3: You are correct, we have rectified the error. Thank you.

Point 4: L107. You say that B. bassiana has low efficacy for ear pest control, but in the summary you recommend the use of this pathogen. Please clarify.

Response 4: Thank you for your insight. In the discussion, we recommend the use of B. bassiana in combination with mineral oil, rather than using B. bassiana alone. The standalone efficiency of B. bassiana is not as high as synthesis pesticides.

Point 5: L126. Planting density "density of 30cm 60cm" What does this mean? Usually densities are written as X thousand plants/ha

Response 5: It has been revised to X thousand plants/ha.

Point 6: Throughout manuscript. Change "0.0667 ha" to 667 m2.  L154. Change to 333 m2.  L169. Delete "per 0.0667 hectare".

Response 6: All corrections have been made.

Point 7: L162 (Table 1) What type of B. thuringiensis was used? var. kurstaki?

Response 7: Currently, we don’t have the classification information for B. thuringiensis. The B. thuringiensis used in the experiment was purchased from Hubei Kangxin Agrochemical Co., Ltd., China. We have added all the pesticide manufacturers' data to Supplementary Table 1.

Point 8: Table 1. What is the dosage and cost unit "he"?? Should this be hectare (ha)?  Line 328 and Table 2 and Fig 4. (+data1 file, sheet 9) Change Decamethrin to Deltamethrin.

Response 8: It should be "ha," and it has been corrected. The other corresponding errors have been rectified.

Point 9: Table 2. What does LoD mean?

Response 9: Limit of Detection (LoD), refers to the lowest concentration of a chemical substance that the instrument can reliably detect under experimental conditions.

Point 10: Fig 3B – how can B. bassiana have a control efficiency of 100% at 0 days post-application? This biological insecticide takes several days to begin killing insects.

Response 10: On the first day after spraying, plants with visible pests and those exhibiting poor growth were excluded and not considered in the subsequent experimental statistics.

Point 11: Fig 3A. Why doesn't the control efficiency of B. bassiana in (A) 78-88% match the control efficiency in (B)? Even if you average the points in B you get higher values than shown in A.

Response 11: In fact, these two experiments were completed in two years. A is the data of 2019, and B is the data of 2021, which are not closely related. It has been marked in the manuscript.

Point 12: L454. "We found that thiamethoxam, may have little to no activity on lepidopteran larvae" – where is your evidence for this assertion? You did not test this.

Response 12: Thank you for your comment. We only guessed based on our experimental results, because Virtako (Thiamethoxam + Chlorantraniliprole) and Chlorantraniliprole have little difference in treatment effect. In fact, Thiamethoxam is mainly used to control predatory pests such as aphids. Future experimental results are indeed needed.

Point 13: L477. What do you mean by "buffer conditions"? Please reword or clarify in text.

Response 13: We have made a more detailed explanation in the original text to prevent misunderstandings.

Point 14: L484. "The findings revealed that most biopesticides are more costly than synthetic insecticides." According to Fig 4. this is not correct. The average cost of the five best chemical insecticides was US$22.9/ha according to my calculations, whereas the five best biopesticides had a cost of US17.9/ha. 

Response 14: Thank you for your careful calculation. We also think that this statement is not accurate. We have deleted this sentence and made modifications.

Point 15: Supplemental material - data1.xls file, sheet 9 – change $/acre values to $/hectare.

Response 15: We have modified it.

Thank you for revising our manuscript so carefully. We appreciate and cherish it very much!